# The phylogeography and incidence of multi-drug resistant typhoid fever in sub-Saharan Africa

Se Eun Park [iD] et al.[#]

There is paucity of data regarding the geographical distribution, incidence, and phylogenetics of multi-drug resistant (MDR) *Salmonella* Typhi in sub-Saharan Africa. Here we present a phylogenetic reconstruction of whole genome sequenced 249 contemporaneous *S.* Typhi isolated between 2008-2015 in 11 sub-Saharan African countries, in context of the 2,057 global *S.* Typhi genomic framework. Despite the broad genetic diversity, the majority of organisms (225/249; 90%) belong to only three genotypes, 4.3.1 (H58) (99/249; 40%), 3.1.1 (97/249; 39%), and 2.3.2 (29/249; 12%). Genotypes 4.3.1 and 3.1.1 are confined within East and West Africa, respectively. MDR phenotype is found in over 50% of organisms restricted within these dominant genotypes. High incidences of MDR *S.* Typhi are calculated in locations with a high burden of typhoid, specifically in children aged <15 years. Anti-microbial stewardship, MDR surveillance, and the introduction of typhoid conjugate vaccines will be critical for the control of MDR typhoid in Africa.

Typhoid fever is a systemic infection primarily caused by the bacterium *Salmonella enterica* serovar Typhi (*S.* Typhi). The organism only infects humans, with the disease being contracted by the ingestion of bacteria through contaminated food or water. The vast majority of the global burden of disease (21.7 million estimated cases annually with 217,000 fatalities)[1] is thought to arise in urban areas in low-middle income countries (LMICs) in South and Southeast Asia, but more recent data have shown a substantial burden of disease in urban and rural areas of sub-Saharan Africa[2]. Between 2010 and 2014, the Typhoid Fever Surveillance in Africa Programme (TSAP) conducted population-based surveillance for typhoid fever in thirteen sites in ten sub-Saharan African countries[3]. The TSAP study, which recruited 13,431 febrile patients, isolated 135 *S.* Typhi from nine countries and found notably high incidences of typhoid fever in Burkina Faso, Ghana, and Kenya[2].

Many antimicrobials remain effective for the treatment of typhoid fever. However, *S.* Typhi that exhibit resistance to empirical antimicrobials hamper successful therapy[4]. The phenomenon of antimicrobial resistance (AMR) in *S.* Typhi has been well described, and resistance to the traditional first-line antimicrobials, ampicillin, chloramphenicol, and trimethoprim-sulfamethoxazole (co-trimoxazole), were associated with large outbreaks in Asia in the 1980s and 1990s[5,6]. The emergence of resistance to these first-line antimicrobials in Asia, which was dominated by the H58 genotype (now renamed 4.3.1)[7,8], led to a change in typhoid treatment guidelines, with fluoroquinolones becoming the empirical choice for MDR infections[9,10]. However, this shift towards the more common use of fluoroquinolones was inevitably followed by a decline in susceptibility to this group of antimicrobials[4,11].

Recent phylogenetic analyses further suggest that the multi-drug resistant (MDR) *S.* Typhi genotype 4.3.1 dominates and circulates across Southeast (lineage I: Vietnam, Cambodia, and Laos) and South Asia (lineage II: mostly India with clusters in Nepal and Pakistan)[12]. Additionally, these 4.3.1 *S.* Typhi have transferred from South Asia into Eastern and Southern Africa (lineages I and II; Kenya, Tanzania, Malawi, South Africa)[12–14]. The characteristics of 4.3.1 *S.* Typhi define this genotype as a key driving force in global MDR *S.* Typhi, as intercontinental transmission, regional circulation, and multiple localised outbreaks over the last three decades are distinct from the evolutionary trends and population structure of other extent *S.* Typhi genotypes[12,15]. Despite the known circulation of 4.3.1 *S.* Typhi in sub-Saharan Africa, there is a paucity of data regarding the geographical distribution of AMR genotypes (MDR and reduced fluoroquinolone susceptibility), their phylogenetic structure, and the incidence of MDR typhoid fever across the African continent. Here, we aimed to investigate the phylogeography and incidence of MDR *S.* Typhi across sub-Saharan Africa, utilizing organisms generated through the TSAP initiative[2,3] and additional typhoid fever studies conducted in Ghana, Uganda, and The Gambia.

## Results

### Geographical distribution of *S.* Typhi genotypes in Africa.
Phylogenetic analysis of 249 contemporary African *S.* Typhi genome sequences combined with 2,057 existing *S.* Typhi genome sequences (including 504 from Africa) permitted a visualisation of these new African isolates within a global *S.* Typhi genomic framework (Fig. 1). The primary observation was that these 249 contemporary African *S.* Typhi sequences were distributed throughout this framework, with multiple lineages found to be circulating simultaneously across sub-Saharan Africa in the last decade. With TSAP providing expansive sampling across the continent, we observed a substantial degree of genetic diversity,

with 12 different *S.* Typhi genotypes represented in 11 different typhoid endemic countries (Fig. 2). This distribution of genotypes ranged from single organisms in particular countries (for example: The Gambia, Kenya, and Uganda) to numerous closely related organism clusters isolated in several countries (Supplementary Table 1).

Despite the apparent broad genetic diversity in the circulating *S.* Typhi population, the majority of the recently isolated organisms (225/249; 90%) belonged to only three genotypes, 4.3.1 (H58) (99/249; 40%), 3.1.1 (97/249; 39%), and 2.3.2 (29/249; 12%). Organisms belonging to genotype 4.3.1 were found only in East Africa, comprising 100% of the *S.* Typhi isolates from Kenya (59/59) and Uganda (30/30), and 91% (10/11) of the isolates from Tanzania (Fig. 1 and Fig. 2). Conversely, all of the organisms belonging to genotype 3.1.1 were found only in West African sites, constituting 88% (89/101) and 57% (8/14) of the *S.* Typhi organisms sequenced from Ghana and Burkina Faso, respectively. Organisms belonging to genotype 2.3.2 were found only in the West African countries of Burkina Faso, The Gambia, Ghana, Guinea-Bissau, and Senegal (Fig. 2 and Supplementary Table 1).

### MDR phenotypes restricted to dominant *S.* Typhi genotypes.
The MDR phenotype was prevalent across isolates from the 11 sampled countries, with 129/249 (52%) of all isolates exhibiting the classical *S.* Typhi MDR phenotypic profile of resistance against ampicillin, chloramphenicol, and co-trimoxazole. MDR organisms were widely distributed in both East and West Africa, and isolated in Ghana (68/101; 67%), Kenya (50/59; 85%), Tanzania (4/11; 36%), and Uganda (7/30; 23%). No MDR organisms were identified in Burkina Faso (0/14), Ethiopia (0/2), The Gambia (0/11), Guinea-Bissau (0/3), Madagascar (0/8), Senegal (0/8), or South Africa (0/2), and none of the organisms in these countries were genotype 4.3.1 or 3.1.1 except for Burkina Faso (8/14; genotype 3.1.1) (Fig. 2 and Table 1).

Saliently, MDR phenotypes were confined entirely within the dominant circulating genotypes in East (4.3.1) and West Africa (3.1.1). Overall, 70% (68/97) of 3.1.1 *S.* Typhi and 62% (61/99) of 4.3.1 *S.* Typhi were MDR (Supplementary Table 1). Further investigation revealed distinct origins of these MDR *S.* Typhi genotypes in each region. These contemporary genome sequences were compared to the existing global framework for *S.* Typhi 4.3.1 using a maximum likelihood phylogeny[12] (Fig. 3a). Our Kenyan MDR 4.3.1 organisms (2012–2013) belonged to two distinct clades, one in lineage I and the other in lineage II, indicative of the two distinct introductions from South Asia, as identified in an earlier global study[12], followed by the establishment of local populations. The Tanzanian MDR 4.3.1 organisms (2011–2012) clustered within each of these Kenyan clades, providing evidence of historical transfer of 4.3.1 *S.* Typhi from Kenya into Tanzania; ongoing local expansion was evident in the lineage I group only. The MDR 4.3.1 *S.* Typhi isolated in Uganda in 2015 formed a monophyletic clade within lineage II that was not closely related to the Kenyan or Tanzanian lineage II organisms, and were characterised by extremely narrow genetic diversity (mean pairwise genetic distance of 1 SNP), indicative of a recent population expansion or an outbreak[16]. This Ugandan MDR cluster was nested within a clade of 4.3.1 MDR *S.* Typhi organisms isolated in South Asia between 2007 and 2011, consistent with a third importation of MDR *S.* Typhi into East Africa from South Asia (Fig. 3a).

In contrast, the 3.1.1 MDR *S.* Typhi from Ghana (68 isolates) represented a population that was found only in West Africa, with the resulting phylogeny showing no evidence for intercontinental transmission as observed for 4.3.1 (Table 1). Rather, 3.1.1 *S.* Typhi could be better defined as a repeating pattern of

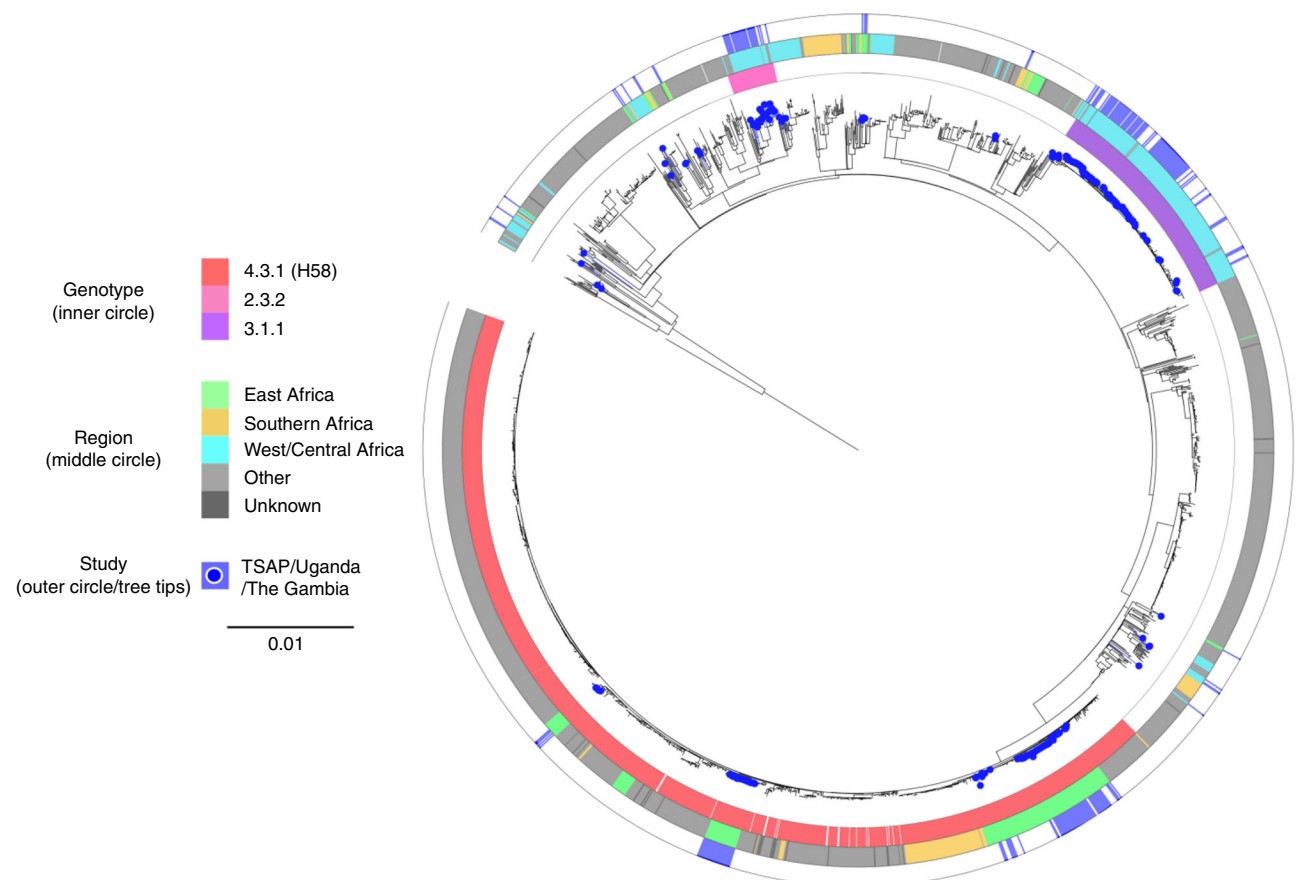

**Fig. 1** The phylogenetic context of *Salmonella* Typhi isolated in sub-Saharan Africa. Maximum likelihood tree outlining the phylogenetic structure of 249 *S.* Typhi isolates unique to this study (highlighted by the blue points) combined with 2,057 global *S.* Typhi isolates. The tree is adjacent to three concentric circles highlighting associated metadata. The inner most circle represents the three most predominant genotypes (colour coded according to top of key), the middle circle represents the geographical sub-regions of Africa from where the *S.* Typhi organisms were isolated (colour coded according to top of key), and the outer circle (blue) again highlights the organisms unique to this study. The scale bar indicates the number of substitutions per variable site

small country specific population expansions with organisms being regularly transferred between countries (Fig. 3b). Phylogeographical reconstruction has not previously been performed for *S.* Typhi 3.1.1, therefore we conducted a Bayesian spatio-temporal phylodynamics analysis for the subclade using BEAST2 (Fig. 3b). The results suggest that Ghana was the most likely recent source of this 3.1.1 *S.* Typhi population (posterior probability = 0.66) which emerged de novo, and the corresponding source of three major clusters, which then radiated into other nearby countries on multiple occasions. Notably, Ghanaian *S.* Typhi appear to have been the probable origin of 3.1.1 *S.* Typhi in Burkina Faso on at least two separate occasions. Furthermore, existing whole genome sequences of 131 *S.* Typhi from Nigeria, including two isolates from travellers returning to the United Kingdom from Nigeria, demonstrated that 3.1.1 *S.* Typhi has been introduced into Nigeria from Ghana on at least two separate occasions. One of these events, estimated to be between 2010 and 2011, formed a major population expansion encompassing the majority (76/86; 88%) of the isolates from Nigeria.

**Geographically distinct *S.* Typhi IncHI1 MDR plasmids**. We next investigated the genetic mechanisms associated with the MDR phenotypes by inferring AMR gene content in the 249 contemporaneous African *S.* Typhi genome sequences. Across the dataset we identified genes encoding resistance to aminoglyco-sides (*aph(3″)-Ib*, *aph(6)-Id*, and *ant(3″)-Ia*), ampicillin (OXA-1

and TEM-95/-93), chloramphenicol (*catA1*), trimethoprim (*dfrA7*, *dfrA14* and *dfrA15*), sulfonamides (*sul1* and *sul2*), and tetracycline (*tet(A)* and *tet(D)*). Most AMR genes were associated with IncHI1 plasmids. However, the two MDR *S.* Typhi geno-types were associated with distinct plasmid lineages. The 3.1.1 MDR *S.* Typhi from Ghana (68 isolates) carried IncHI1 MDR plasmids of plasmid sequence type (PST) 2a, whilst the 4.3.1 MDR *S.* Typhi from Kenya (50 isolates) and Uganda (7 isolates), respectively carried ST6 and ST6a IncHI1 MDR plasmids. Minor differences in the specific AMR genes were also evident between these plasmid types (Fig. 4). For example, the class I integron cassette contained *dfrA15* in the West African/PST 2a plasmid and *dfrA7* in East African/PST 6/6a plasmids, the latter plasmids also contained *sul2* and *tet(D)* which were absent from the West African isolates.

Outliers included: non-MDR *S.* Typhi isolates from Burkina Faso (genotype 2.2) with an IncX1 plasmid containing no resistance genes and Ghana (genotype 3.1.1) with an IncN plasmid displaying resistance against aminoglycosides (*aph(6)-Id*), ampicillin (TEM-95/-93), trimethoprim (*dfrA14*), and sulfonamides (*sul2*) and 5 non-MDR *S.* Typhi isolates (genotype 4.3.1) from Tanzania with IncFIB plasmid carrying resistant genes *aph(3″)-Ib*, *aph(6)-Id*), TEM-95/-93, *dfrA14*, *sul2*; and an MDR *S.* Typhi isolate (genotype 4.3.1) from Kenya with an IncHI1 and IncQ1 plasmid associated with resistance genes against aminoglycosides (*aph(3″)-Ib*, *aph(6)-Id*), ampicillin (TEM-95/-93), chloramphenicol (*catA1*), trimethoprim (*dfrA7*),

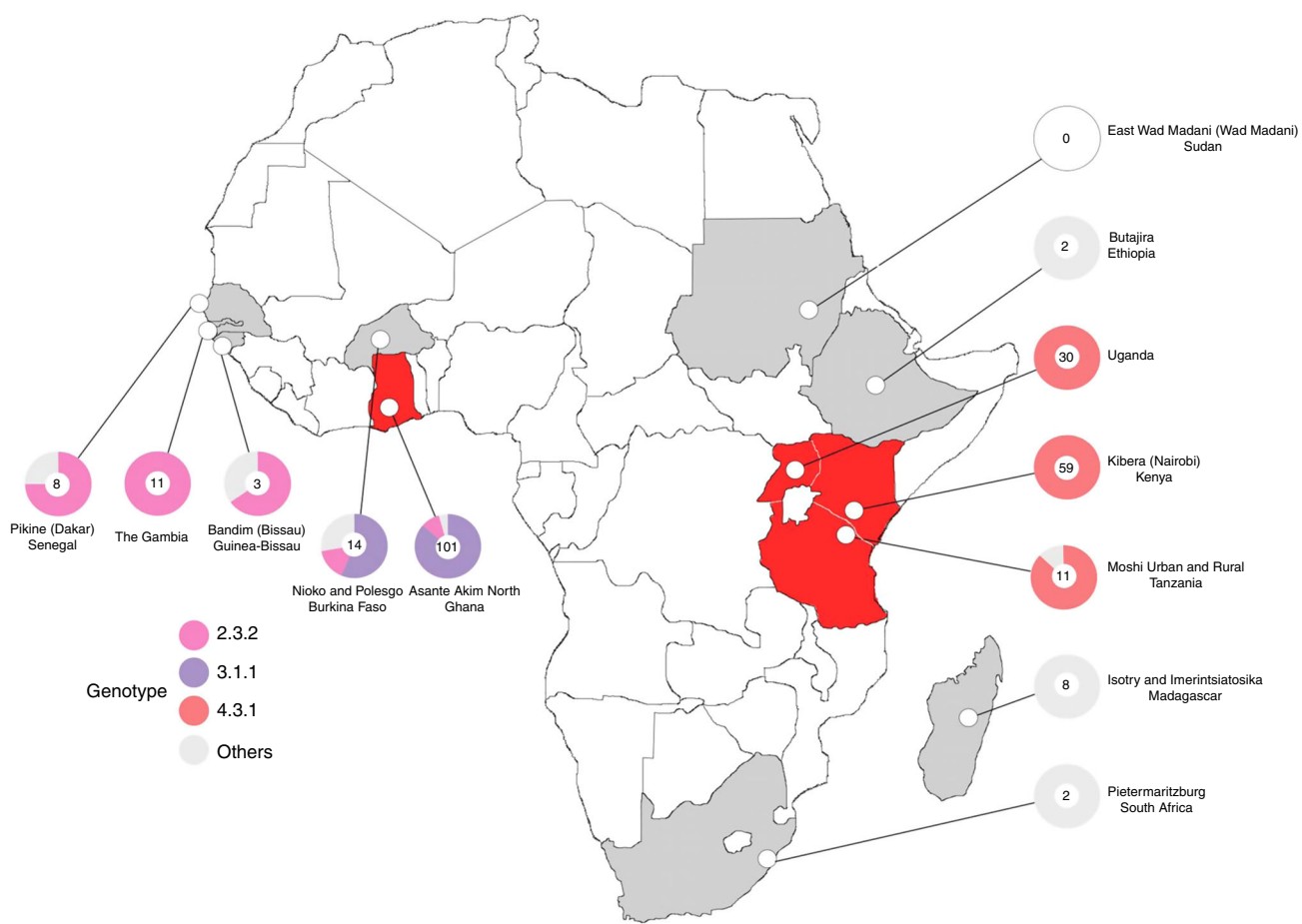

**Fig. 2** The distribution of multi-drug resistant *Salmonella* Typhi isolated in Africa. Map of the African continent showing the locations of the field sites from where the *S*. Typhi organisms were isolated for this study. Countries in which multi-drug resistant (MDR) *S*. Typhi were isolated are coloured in red, countries in which MDR *S*. Typhi were not isolated are coloured in grey. Pie charts correspond with the proportion of the main genotypes isolated (see key), with the number of isolates from each location in the centre

sulfonamide (*sul1* and *sul2*), and tetracycline (*tet(A)* and *tet(D)*) (Fig. 4). Additionally, none of the four MDR organisms from Tanzania possessed a detectable plasmid backbone. Using Bandage to investigate the location of MDR cassettes, we found that these isolates carried multiple resistance genes (*aph(3″)-Ib, aph(6)-Id*), TEM-95/-93, *catA1, dfrA7, sul1, sul2*) on a 24-kb composite chromosomal transposon (Tn2670-like element) inserted between coding sequences STY3618 and STY3619[12].

In total, 16% (39/249) of the contemporaneous African *S*. Typhi exhibited reduced susceptibility against ciprofloxacin (9 from Kenya and 30 from Uganda). The Kenyan organisms exhibited the common mutation associated with reduced susceptibility to fluoroquinolones in *S*. Typhi, a substitution from serine to phenylalanine at codon 83 (Ser83Phe) in *gyrA*. The Ugandan organisms harboured an alternative serine to tyrosine *gyrA* mutation also at codon 83 (Ser83Tyr) (Table 1).

**The incidence of MDR typhoid fever in African countries**. We lastly calculated the incidence of MDR typhoid fever in specific age groups in countries where MDR *S*. Typhi was isolated: Ghana, Kenya, and Tanzania (Table 2). The incidence of MDR *S*. Typhi exceeded 100/100,000-person years of observation (PYO) in specific age groups in Ghana (<15 years: 414/100,000 PYO; 95% confidence interval [CI], 333–515) and Kenya (<15 years: 398/ 100,000 PYO; 95% CI, 291–545), in all age groups in Kenya (263/ 100,000 PYO; 95% CI, 199–347) and in the urban site in Tanzania (103/100,000 PYO; 95% CI, 61–173). While Burkina Faso had a

high overall incidence of typhoid[2], no MDR *S*. Typhi were detected; 2/14 isolates were resistant to chloramphenicol and co-trimoxazole. The highest incidence of MDR *S*. Typhi in a specific age group in a single location was in children aged 2–4 years in Ghana (747/100,000 PYO; 95% CI, 491–1135), followed by 5–14 year olds in Kenya (507/100,000 PYO; 95% CI, 352–729). In Kenya and Ghana, the only TSAP sites where *S*. Typhi were isolated from infants (aged 0–1 years), the incidences of MDR *S*. Typhi in this age group were 148 (95% CI, 48–458) and 60 (95% CI, 17–210) per 100,000 PYO, respectively. Generally, the incidence of MDR *S*. Typhi was substantially higher in children <15 years than in adults. An exception was in Tanzania, where MDR *S*. Typhi occurred in higher incidences in those aged ≥15 years than in young children (Table 2).

## Discussion

Here we present a contemporary dataset of *S*. Typhi genome sequences and AMR data from across sub-Saharan Africa generated through a major population-based surveillance study with data augmented from further locations. We exploited these data to assess the circulation of MDR *S*. Typhi genotypes and to calculate the incidence of MDR typhoid infections across the continent. Our results have major implications for the use of empirical antimicrobials for treating febrile disease of presumed bacterial origin and future intervention measures for controlling typhoid in Africa.

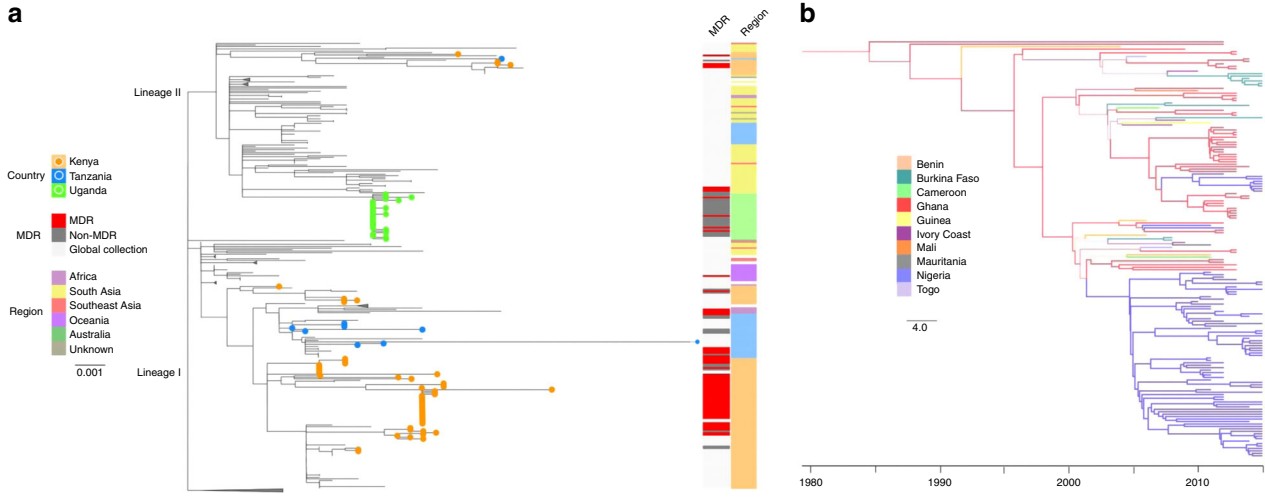

**Fig. 3** The phylogenetic structures of the major *Salmonella* Typhi genotypes in sub-Saharan Africa. **a** Maximum likelihood tree of genotype 4.3.1 *S.* Typhi isolates from this study in the context of other global genotype 4.3.1 *S.* Typhi isolates; the two distinct sub-lineages are labeled at the base of the tree. 4.3.1 *S.* Typhi isolates from this study (Kenya, Tanzania, and Uganda) are highlighted in corresponding coloured branches and circles at the tip of each tree. The first coloured bar shows the MDR phenotypes of study isolates. The second coloured bar outlines the continents and African regions where 4.3.1 *S.* Typhi have been detected. Scale bar indicates the number of substitutions per variable site; nodes of the tree have been collapsed for better visualization. **b** Maximum clade credibility tree (reconstructed using BEAST2) of genotype 3.1.1 *S.* Typhi isolates from this study in the context of other global genotype 3.1.1 *S.* Typhi isolates. Tree shows a phylogeographical reconstruction of genotype 3.1.1 *S.* Typhi isolates in West Africa. Branches are weighted by the support for the location changes; thicker branches have higher support. Branches and nodes are coloured according to the location that had the highest posterior probability values for some nodes of the tree. The scale bar indicates the number of substitutions per variable site per year

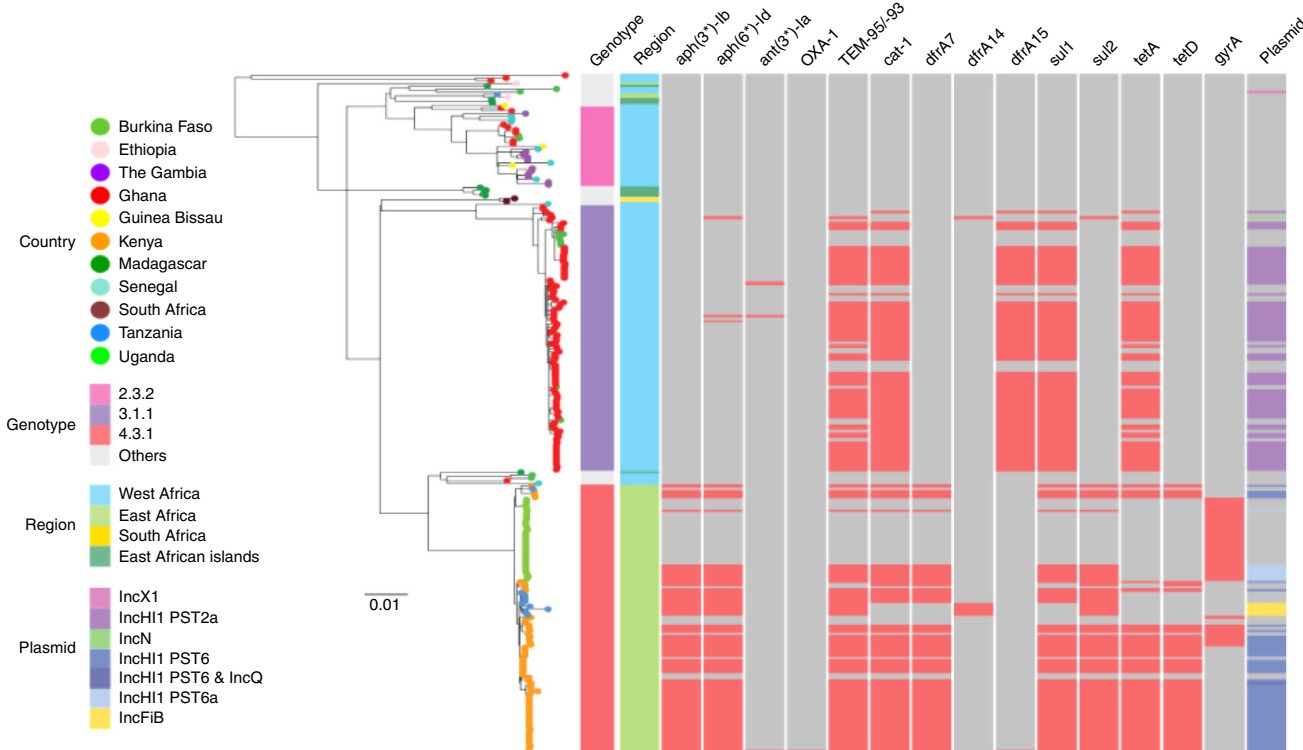

**Fig. 4** The antimicrobial gene distribution within sub-Saharan African *Salmonella* Typhi. Maximum likelihood phylogenetic tree of 249 *S.* Typhi isolates from this study with corresponding metadata including genotype, location, antimicrobial resistance genes (AMR), and plasmids (see keys). Countries where *S.* Typhi isolates were isolated are highlighted by coloured circles at the tip of the branches. The three major genotypes and sub-regions of the Africa continent are shown by the coloured bars; present AMR genes are shown in red. The scale bar indicates the number of substitutions per variable site

## Table 1 Genotypes of MDR[a] *S. Typhi* and *gyrA* in four countries[b]

| Country (n, all S. Typhi per country) | MDR S. Typhi (n) | MDR S. Typhi (%/out of all S. Typhi per country) | MDR S. Typhi Genotype | Non-susceptible to fluoroquinolones (n, gyrA)[c] |
|---|---|---|---|---|
| Ghana (101) | 68 | 67 | 3.1.1 | 0 |
| Kenya (59) | 50 | 85 | 4.3.1 (H58) | 9 (Ser83Phe) |
| Tanzania (11) | 4 | 36 | 4.3.1 (H58) | 0 |
| Uganda (30) | 7 | 23 | 4.3.1 (H58) | 30 (Ser83Tyr) |
| Total 201 S. Typhi from 4 countries (out of 249 S. Typhi from 11 countries[d]) | 129 | 64% of 201  52% of 249 | | |

[a]MDR definition used for the analysis: presence of resistant genes for at least one agent in all three antimicrobial categories of ampicillin/amoxicillin (beta-lactamase: OXA-1, TEM-95/-93) AND chloramphenicol (*catA1*), AND trimethoprim-sulfamethoxazole (sulfonamide (*sul1*, *sul2*) and trimethoprim (*dfrA7, dfrA14, dfrA15*))
[b]Four countries with MDR *S. Typhi* organisms: Ghana, Kenya, Tanzania, and Uganda
[c]Out of total 249 *S. Typhi* isolates yielded from this study in 11 countries in sub-Saharan Africa, total 39 isolates were non-susceptible to fluoroquinolone (ciprofloxacin and nalidixic acid (*gyrA*)): 9 of 39 isolates were from Kenya, of which 7 were MDR *S. Typhi*; and all 30 isolates from Uganda were non-susceptible to fluoroquinolones, of which 7 were MDR *S. Typhi*. These 39 organisms exhibited the mutations at codon 83 of *gyrA*; serine (TCC) to phenylalanine (TTC) for all 9 isolates from Kenya (Ser83Phe) and serine (TCC) to TAC (tyrosine) for all 30 Uganda isolates (Ser83Tyr)
[d]No MDR *S. Typhi* from Burkina Faso (14; genotypes 2.2 (2 isolates), 2.3.2 (2 isolates), 3.1.1 (8 isolates), and 4.1.1 (2 isolates)), Ethiopia (2; genotypes 1.2 (1 isolate) and 2.2.2 (1 isolate)), Gambia (11 isolates, all genotype 2.3.2), Guinea-Bissau (3; genotypes 2.3.2 (2 isolates) and 2.3.1 (1 isolate)), Madagascar (8; genotypes 2.5 (4 isolates), 2.2 (3 isolates), and 4.1 (1 isolate)), Senegal (8; genotypes 2.3.2 (6 isolates), 3.1 (1 isolate), and 4.1 (1 isolate)), and South Africa (2; all genotypes 3.1.1)

## Table 2 The incidence of MDR typhoid fever in sub-Saharan Africa[a]

| Country | Age group in years | PYO estimation[b] | | | | Recruitment proportion[b] | Genome-sequenced S. Typhi cases[c] | Crude MDR S. Typhi cases | Crude MDR S. Typhi incidence per 100,000 PYO | Adjusted MDR S. Typhi cases | Adjusted MDR S. Typhi incidence per 100,000 PYO (95% CI)[d] |
|---|---|---|---|---|---|---|---|---|---|---|---|
| | | Proportion of catchment population visiting study facility in case of fever (95% CI) | Catchment population | Catchment population adjusted by health-seeking behavior | PYO | | | | | | |
| *Ghana*[e] AAN | 0-1 | 16% (14-18) | 11222 | 1760 | 4080 | 41% | 1 | 1 | 25 | 2 | 60 (17-210) |
| | 2-4 | 16% (13-18) | 8086 | 1268 | 2940 | 41% | 17 | 12 | 306 | 22 | 747 (491-1135) |
| | 0-4 | n.a. | n.a. | n.a. | n.a. | n.a. | 18 | 13 | n.a. | n.a. | n.a. |
| | 5-14 | 16% (15-17) | 34439 | 5415 | 12554 | 623/1657 (38%) | 23 | 16 | 96 | 24 | 252 (177-357) |
| | <15 | n.a. | 53747 | 8443 | 19574 | n.a. | 41 | 29 | 97 | 81 | 414 (333-515) |
| | ≥15 | n.a. | n.a. | n.a. | n.a. | n.a. | 22 | 16 | n.a. | n.a. | n.a. |
| | Non_TSAP[e] | n.a. | n.a. | n.a. | n.a. | n.a. | 38 | 23 | n.a. | n.a. | n.a. |
| | All | n.a. | n.a. | n.a. | n.a. | n.a. | 101 | 68 | n.a. | n.a. | n.a. |
| *Kenya* Kibera | 0-1 | 42% (38-47) | 3467 | 1456 | 2031 | 99/99 (100%) | 5 | 3 | 148 | 3 | 148 (48-458) |
| | 2-4 | 39% (36-43) | 3070 | 1197 | 2039 | 312/312 (100%) | 11 | 7 | 343 | 7 | 343 (164-720) |
| | 5-14 | 43% (39-47) | 7514 | 3231 | 5722 | 539/539 (100%) | 32 | 29 | 507 | 29 | 507 (352-729) |
| | <15 | n.a. | 14051 | 5884 | 9792 | | 48 | 39 | 398 | 39 | 398 (291-545) |
| | ≥15 | 35% (32-38) | 15263 | 5342 | 9228 | 301/301 (100%) | 11 | 11 | 119 | 11 | 119 (66-215) |
| | All | n.a. | 29314 | 11227 | 19020 | n.a. | 59 | 50 | 263 | 50 | 263 (199-347) |
| *Tanzania*[f] Moshi Rural | 0-1 | 4% (0-11) | 24289 | 390 | 693 | 79% | 0 | 0 | 0 | 0 | 0 |
| | 2-4 | 2% (0-4) | 25281 | 406 | 721 | 79% | 0 | 0 | 0 | 0 | 0 |
| | 5-14 | 13% (10-16) | 118219 | 15487 | 27508 | 79% | 1 (2)[f] | 0 | 0 | 0 | 0 |
| | <15 | n.a. | 167789 | 16283 | 28922 | n.a. | 1 (2)[f] | 0 | 0 | 0 | 0 |
| | ≥15 | 2% (1-2) | 298948 | 5172 | 9186 | 79% | 2 (4)[f] | 0 | 0 | 0 | 0 |
| | All | n.a. | 466737 | 21454 | 38108 | n.a. | 3 (6)[f] | 0 | 0 | 0 | 0 |
| Moshi Urban | 0-1 | 7% (0-19) | 10406 | 335 | 595 | 79% | 0 | 0 | 0 | 0 | 0 |
| | 2-4 | 2% (0-6) | 10831 | 348 | 618 | 79% | 0 | 0 | 0 | 0 | 0 |
| | 5-14 | 13% (8-19) | 37309 | 4850 | 8615 | 79% | 3 (9)[f] | 2 (7)[f] | 12 (81)[f] | 1 (9)[f] | 15 (3-84) (103 (54-199))[f] |
| | <15 | n.a. | 58546 | 5533 | 9828 | n.a. | 3 (9)[f] | 2 (7)[f] | 10 (71)[f] | 1 (9)[f] | 10 (1-72) (91 (47-175))[f] |
| | ≥15 | n.a. | 125746 | 2138 | 3796 | 79% | 4 (8)[f] | 2 (4)[f] | 53 (105)[f] | 3 (5)[f] | 67 (19-229) (133 (56-319))[f] |
| | All | n.a. | 184292 | 7671 | 13626 | n.a. | 7 (17)[f] | 4 (11)[f] | 29 (81)[f] | 4 (14)[f] | 29 (11-78) (103 (61-173))[f] |

[a]The TSAP study has data from total 10 countries, of which 9 countries (Burkina Faso, Ethiopia, Ghana, Guinea-Bissau, Kenya, Madagascar, Senegal, South Africa, and Tanzania in alphabetical order) found blood culture confirmed *S. Typhi* isolates circulating in the respective sites. These *S. Typhi* isolates have been whole-genome sequenced for detection of multidrug resistant (MDR) genes. In addition, *S. Typhi* isolates yielded from 2 other surveillance activities in Uganda and The Gambia have been added to this analysis. Of these 11 countries, *S. Typhi* isolates with MDR genes were detected in Ghana from West Africa and Kenya, Tanzania, and Uganda from East Africa. Incidence of MDR *S. Typhi* in Uganda could not be estimated due to insufficient data on age stratification of patients, catchment population, healthcare seeking behavior and recruitment proportion, which were applied uniformly for the analysis presented in this table for Ghana, Kenya, and Tanzania
[b]PYO estimation and recruitment proportion have been published in detail in the TSAP typhoid burden paper (Marks et al, Lancet Global Health, 2017)
[c]Genome sequenced *S. Typhi* case numbers in this table may not exactly match the crude *S. Typhi* case numbers reported in the TSAP typhoid burden paper (Marks et al) due to few sequencing failures
[d]Adjusted incidence rates per 100,000 PYO (95% CI): adjustments for case recruitment and error factors
[e]Ghana samples include non-TSAP projects as outlined in the Supplementary Table 2. AAN: Asante Akim North (Supplementary Table 2)
[f]Tanzania: Enrolment algorithm has been applied to the crude MDR *S. Typhi* case numbers, that is: recruitment by every 5th patient if enrolled before Nov 11th 2011 and every 2nd patient if enrolled after then. 1 isolate from Tanzania, which was from outside the study catchment area (Supplementary Table 2: "Moshi Other") is not included in this incidence table due to the insufficient background data required as mentioned in this footnote

Despite the broad genetic diversity observed within the continental *S.* Typhi population, we identified only three principal *S.* Typhi genotypes. These genotypes were geographically limited to East (genotype 4.3.1) and West (genotypes 3.1.1 and 2.3.2) Africa. MDR *S.* Typhi in Africa is currently dominated by genotypes 4.3.1 and 3.1.1. *S.* Typhi 4.3.1 has been previously reported to circulate only in East Africa on the African continent[12–14], with 3.1.1 dominating in Nigeria and circulating amongst neighbouring countries in West Africa[17]. After the likely importation from South Asia within the last 20 years, the extant population of *S.* Typhi 4.3.1 in Kenya, Tanzania, and Uganda has been formed through multiple introductions from South Asia followed by local expansions. Conversely, *S.* Typhi 3.1.1, which were isolated in Ghana, Burkina Faso, and Nigeria, do not appear to have recent ancestral roots in Asia, but have undergone localised microevolution within West Africa in recent decades. We speculate that these organisms have been transferred, maintained, and selected through the sustained movement of people and antimicrobial usage in West Africa. The MDR 4.3.1 *S.* Typhi from Kenya and Uganda also commonly exhibited mutations in *gyrA*, associated with reduced susceptibility to fluoroquinolones, which has also been reported in Africa in recent years. Conversely, no *gyrA* mutations were found in the MDR *S.* Typhi 3.1.1 from Ghana. These data mirror recent reports from Nigeria[17], and suggest that first-line antimicrobial agents (ampicillin, chloramphenicol, and co-trimoxazole) for the treatment of febrile diseases are still in common use in West Africa.

The acquisition of an MDR phenotype in *S.* Typhi is typically associated with IncHI1 plasmids, which have long been considered the main vehicle for resistance to first-line antimicrobials in *S.* Typhi[8]. The distinct MDR lineages of *S.* Typhi found in West and East Africa, each associated with a distinct IncHI1 plasmid sequence type, suggest that *S.* Typhi and its AMR plasmids have not been transferred laterally across the continent. This may be because genotype 4.3.1 MDR *S.* Typhi has not been circulating for a sufficient period in Africa to reach the West African region. Furthermore, the four MDR *S.* Typhi isolates from Tanzania did not harbour plasmid-associated sequences, suggesting that these AMR genes are inserted into the chromosome, as has been observed previously in Asia[12,18,19] and Zambia[20]. The integration of AMR genes into the *S.* Typhi chromosome is a worrying development, as it provides a mechanism for stable vertical transmission of the MDR phenotype without the potential fitness deficit associated with maintaining large plasmids, increasing the likelihood that MDR will be sustained during the ongoing spread of related *S.* Typhi across East Africa.

Here we identified specific populations that are most at risk of MDR typhoid, which particularly warrants a reconsideration of current empirical antimicrobial use for treatment of typhoid. Generally, we found that the site incidences of MDR *S.* Typhi corresponded largely with the overall burden of typhoid in the various study sites[2] (that is, countries with high incidences of typhoid also had high incidences of MDR *S.* Typhi). Consequently, Kenya and Ghana exhibited the highest incidences of MDR typhoid in the sampled countries. Notably, Burkina Faso, which had a high burden of typhoid, had no incidence of MDR *S.* Typhi in comparison to neighbouring Ghana. Further, we found that children aged <15 years, the highest at-risk age group for typhoid in Africa, also generally exhibited the highest incidence rates of MDR *S.* Typhi infections. This age distribution of typhoid caused by MDR *S.* Typhi was not consistent across the continent, as those aged >15 years in Tanzania exhibited a higher incidence of MDR *S.* Typhi than younger children. Alternatively, some sites with a high burden of typhoid in specific age groups had no MDR infections. We suggest that this distribution is likely to mirror access to, and the generic usage of, specific antimicrobial agents in

these locations and age groups, warranting the need for continued country/site-specific surveillance, review of local treatment policies, and the collection of antimicrobial usage data.

The incidence of MDR typhoid varied dramatically between settings and also between age groups in some individual locations. This discrepancy may be due to differing exposures to antimicrobials in different settings and age groups, which could lead to differential selective pressures in local circulating bacterial populations. Our data additionally indicate that AMR/MDR *S.* Typhi are not only spread through local population movements in East and West Africa but can also arise de novo. This phenomenon can be observed within the microevolution and expansion of 3.1.1 MDR *S.* Typhi in West Africa. The AMR genes associated with 4.3.1 MDR *S.* Typhi in East Africa appear to be both plasmid and chromosomally located. This observation, coupled with the acquisition of reduced susceptibility to fluoroquinolones, transmission between East African countries, and the importation of organisms from South Asia, raises further concerns regarding the progression of drug resistant *S.* Typhi in Africa. 4.3.1 *S.* Typhi has spread successfully cross South Asia and become increasingly resistant to ciprofloxacin, making treatment options more limited[4]. The pervasiveness of AMR in 4.3.1 *S.* Typhi in South Asia has been recently highlighted by an outbreak of a ceftriaxone-resistant 4.3.1 *S.* Typhi in Hyderabad, Pakistan, which appears to be resistant to commonly available antimicrobial classes[21]. We predict that new AMR phenotypes that emerge in 4.3.1 *S.* Typhi in Asia can be periodically introduced into East Africa. Further, the emergence of MDR *S.* Typhi 4.3.1 in South Africa suggests possible spread from East Africa to Southern Africa through human population movement, however this notion requires further investigation[22].

This study highlights locations in sub-Saharan Africa where MDR typhoid is prevalent and where future activities to control its spread from Asia into Africa and also within Africa could be focused. In addition to continuing disease surveillance and investigating the genomic characteristics and phenotypic profiles of MDR *S.* Typhi, compiling antimicrobial usage data that can be linked with the distribution of AMR/MDR bacterial pathogens across Africa is becoming essential. The World Health Organization (WHO) has prequalified a typhoid conjugate vaccine (TCV) in January 2018 with a recommendation to introduce the vaccine for infants and children older than six months in typhoid endemic countries[23]. Targeted vaccination programs at sites with a high burden of AMR/MDR *S.* Typhi could also be considered and may be informed by the age-stratified MDR disease incidence data presented here. New and potentially highly efficacious *S.* Typhi conjugate vaccines are currently undergoing clinical trials and should become routinely available at the end of this decade[23]. Until these vaccines become available, countries in Africa with endemic typhoid should structure antimicrobial stewardship policies to control MDR *S.* Typhi and develop national roadmaps for their deployment.

## Methods

**Bacterial isolates and antimicrobial susceptibility testing**. Between 2010 and 2014, a population-based surveillance of invasive *Salmonella* infections was conducted in ten sub-Saharan countries (see Supplementary Table 2)[2]. The research methodology including ethics approvals, sampling framework, and calculation of disease incidence of this programme have been previously reported[3]. Briefly, over the TSAP sampling period, blood culture-based surveillance was conducted in defined catchment areas. Cultured isolates were assessed for antimicrobial susceptibilities by the disc diffusion method locally and at a central reference laboratory. TSAP recruited 13,558 patients meeting the study inclusion criteria, of which 127 patients were excluded due to incomplete data. This resulted in 13,431 patients and 135 *S.* Typhi found in 9 countries for analysis[2]. We also included 114 additional *S.* Typhi collected from other studies in Africa (Uganda; 2015, Gambia; 2008-2014 and non-TSAP isolates from Ghana; 2010), resulting in a collection of 249 *S.* Typhi (Supplementary Table 2).

**Genome sequencing and SNP calling**. Genomic DNA from the 249 *S.* Typhi isolates was extracted using the Wizard Genomic DNA Extraction Kit (Promega, Wisconsin, USA). Two µg of genomic DNA from each organism was subjected to indexed-tagged pair-end sequencing on an Illumina Hiseq 2000 platform (Illumina, CA, USA) to generate 100 bp paired-end reads. To identify single nucleotide polymorphisms (SNPs), raw Illumina reads were mapped to the reference sequence of *S.* Typhi CT18 (accession: AL513382) including plasmids pHCM1 (accession: AL513383) and pHCM2 (accession: AL513384), using SMALT version 0.7.4. Candidate SNPs were called against the reference sequence using SAMtools[24] and filtered with a minimal phred quality of 30 and minimum consensus base agreement of 75%. The allele at each locus in each isolate was determined by reference to the consensus base in that genome using SAMtools *mpileup* and removing low confidence alleles with consensus base quality ≤20, read depth ≤5 or a heterozygous base call. SNPs in phage regions, repetitive sequences, or recombinant regions identified previously were excluded[12,25]. We further identified an additional recombinant region from the whole genome alignment produced by SNP-calling isolates using Gubbins[26] and SNPs detected within this region (~20kb from nucleotide 1,439,032-1,459,472) were removed, resulting in a final set of 4,444 chromosomal SNPs. The SNP data were used to assign all isolates to previously defined subclades in the *S.* Typhi genotyping framework[15].

**Phylogenetic analysis**. A maximum likelihood (ML) phylogenetic tree was constructed from the 4,444 SNP alignment using RAxML version 8.2.8 with a generalized time-reversible model and a Gamma distribution to model the site-specific rate variation (GTR+$\Gamma_4$ model)[27]. Branch support for this tree was assessed through a bootstrap analysis with 1,000 pseudo-replicates. To investigate the molecular epidemiology of our African isolates in regional and international context, a secondary ML tree was inferred from a separate alignment of 26,479 SNPs identified across a total of 2,306 *S.* Typhi isolates (249 from this study, 1,830 from the global collection[12], 128 from Nigeria[17], and 99 travel-associated *S.* Typhi organisms isolated in the United Kingdom[15]) using RAxML with GTRGAMMA substitution model and *S.* Paratyphi A sequence data to outgroup root the tree. Branch support for this phylogeny was assessed through a 100 bootstrap pseudo-analysis. Annotation of this global tree was visualized using ITOL[28]. An interactive version of the global phylogeny, with organisms labeled by genotype, country of origin, year of isolation and antimicrobial susceptibility was generated in Microreact[29].

**Evolutionary timescale and phylogeographic patterns**. For genotype 3.1.1 strains, Bayesian phylogenetic analyses was conducted in BEAST2 v2.4.7[30]. The GTR+$\Gamma_4$ substitution model, an uncorrelated lognormal relaxed-clock model, and the exponential-growth coalescent tree prior were used. Three independent analyses were performed with $5\times10^8$ steps, recording samples every $5\times10^4$ steps. We assessed sufficient sampling by combining the three independent runs and verifying that the effective sample size of all parameters was at least 500. To calibrate the molecular clock, we used the sampling year of all sequences. This analysis also included an outgroup sequence (CT18) to ensure a biologically meaningful root location. Our selected molecular clock model and tree prior have been shown to perform well even when the data display low rate variation and constant population size dynamics[31]. This model combination also allows for informal model testing via the coefficient of rate variation and the population growth rate parameters[32,33]. To determine phylogeographic patterns, we considered the country of sampling as a discrete trait in our analysis in BEAST2[34]. A potential shortcoming of this analysis was that it includes a large number of parameters (transition rates between all locations), therefore the output of these analysis may be affected by the prior distribution. We verified that the prior distribution differed from the posterior by comparing the distributions of all transition rates.

An important consideration when using sampling times as calibrations is that the sampling timespan should capture sufficient genetic variation to allow reliable inferences of evolutionary rates and timescales, such that the data have strong temporal structure. We verified the temporal structure in the data by using a root-to-tip regression and a date-randomisation test[35]. We conducted a root-to-tip regression for the outgroup-rooted ML tree using TempEst[36], and obtained a positive value for the slope, an $R^2$ of 0.12, and a *p*-value of $3\times10^{-6}$ (Supplementary Figure 1). For the date-randomisation test we repeated the analysis 20 times while randomising the sampling times. Our expectation was that the randomisations should produce evolutionary rate estimates that were lower and that did not overlap with those obtained with the correct sampling times[37], which was the case for our data (Supplementary Figure 2). Finally, we compared our estimate of the time of origin of the 3.1.1 lineage in BEAST with an independent method, LSD v0.3[38]. LSD and BEAST2 produced congruent estimates of the time of origin of the 3.1.1 lineage (Supplementary Figure 3).

**Antimicrobial resistance gene and plasmid analyses**. ARIBA (Antimicrobial Resistance Identifier by Assembly)[39] and CARD (https://card.mcmaster.ca/home) were used to investigate AMR gene content. ARIBA reported the AMR genes and the quality of assemblies and variants detected between the sequencing reads and the reference sequences, including mutations in the quinolone resistance-determining region (QRDR) of the *gyrA*, *gyrB*, *parC*, and *parE* genes. For plasmid

identification, the sequence reads from each isolate were de novo assembled using the short-read assembler Velvet with parameters optimized by Velvet Optimizer[40,41]. Contigs that were less than 300 bp long were excluded and the assembled contigs were annotated using Prokka[41,42]. Plasmid typing was performed *in silico* using PlasmidFinder[43]. The presence of the IncHI1 plasmid was confirmed by BLASTN searching the assembled sequences in reference to the pHCM1 reference plasmid sequence, and comparative analyses were performed and visualized using ACT[44]. The IncHI1 plasmid sequence type was identified using SRST2 software[45] with the IncHI1 plasmid MLST scheme[46]. To investigate the isolates with MDR phenotype and without plasmid, raw sequences were subjected to de novo genome assembly using SPAdes[47] version 3.11.0, and the resulting assembly graph was visualized in Bandage[48] to inspect the location of AMR genes in the genome.

**Incidence analyses of MDR S. Typhi**. Incidence of MDR *S.* Typhi was estimated per 100,000 person-years of observation (PYO) for MDR *S.* Typhi isolates found in Ghana, Kenya and Tanzania. Statistical methodology used previously to calculate the incidence of *S.* Typhi TSAP isolates[2,3] was applied to calculate MDR *S.* Typhi incidence. Briefly, age-stratified PYO were estimated using available demographic data in HDSS (Health and Demographic Surveillance System) and non-HDSS sites and health-seeking behaviour of randomly selected individuals, representative of the study population, were factored in (denominator). The recruitment proportion was adjusted to the age-stratified crude MDR *S.* Typhi cases (numerator). Adjusted incidence of MDR *S.* Typhi per 100,000 PYO was estimated with 95% CIs using these adjustment factors and crude MDR *S.* Typhi case numbers. The previously established multi-country database (FoxPro software) for TSAP was used for the three countries with MDR *S.* Typhi. The incidence of MDR *S.* Typhi in Uganda could not be measured, as data regarding adjustment factors (healthcare seeking behaviour and recruitment proportion) was unavailable at the time of analysis.

## Data availability

Raw sequence data are available in the European Nucleotide Archive (projects ERP009684, ERP010763, ERP013866). The Microreact interactive phylogeny of the analysed isolates is available at: https://microreact.org/project/HJWBihsvz. SMALT version 0.7.4 used is available at: http://www.sanger.ac.uk/resources/software/smalt/.

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

## Acknowledgements

This study was supported by the Bill & Melinda Gates Foundation (grant: OPPGH5231). The findings and conclusions contained within are our own and do not necessarily reflect positions or policies of the Bill & Melinda Gates Foundation or the US Centers for Disease Control and Prevention, the International Vaccine Institute. The International Vaccine Institute acknowledges its donors, including the Government of the Republic of Korea and the Swedish International Development Cooperation Agency (SIDA). Research infrastructure at the Moshi site was supported by the US National Institutes of Health (R01TW009237; U01 AI062563; R24 TW007988; D43 PA-03-018; U01 AI069484; U01 AI067854; P30 AI064518), and by the UK Biotechnology and Biological Sciences Research Council (BB/J010367). SB is a Sir Henry Dale Fellow, jointly funded by the Wellcome Trust and the Royal Society (100087/Z/12/Z). KEH is supported by a Senior Medical Research Fellowship from the Viertel Foundation of Australia. Z.A.D. was supported by strategic award #106158 from the Wellcome Trust of Great Britain (106158/Z/14/Z). V.K.W. is supported by NIHR and NIHR Cambridge BRC. We thank staff at the TSAP network countries, the Medical Research Council Unit The Gambia, and the Uganda Public Health Emergency Operations Centre, Kampala, Uganda. We are grateful to Soo-Young Kwon for her invaluable administrative support of the project.

## Author contributions

S.E.P., D.P.T., C.B., V.K.W., G.D.P., S.D. and Z.D. contributed in data analysis, interpretation of results under the scientific guidance from S.B. and K.E.H. S.E.P. drafted and edited the paper and S.B. contributed in structuring and editing of the paper. F.M. contributed in project oversight. F.M., S.E.P., V.vK., L.M.C.E., U.P., G.D.P., J.I., H.S-G., J.A.C., R.F.B., Y.A-S., E.O-D., R.R., A.B.S., A.A., N.G., K.H.K., J.M., A.G.S., P.A., H.M.B., J.T.H., J.M.M., L.C., B.O., B.F., N.S., T.J.L.R., T.M.R., L.P.K., E.S., M.T., B.Y., M.A.E.T., A.S., S.P., A.T., A.N., M.B.A., S.V.L., H.J.S., H.J.J., J.F.D., J.K.P., F.K., P.H., J.N.S., T.V., R.D., U.N.I., G.A.M. and S.O. contributed to data acquisition in the field, interpretation of results, and editing of the paper. G.D., J.A.K., A.P., D.M.A., S.A., M.A. and J.D.C. contributed to data interpretation. All authors read and approved the final draft.

## Additional information

**Competing interests:** The authors declare no competing interests.

Se Eun Park [1,2], Duy Thanh Pham[2], Christine Boinett[2,3], Vanessa K. Wong[4,5], Gi Deok Pak[1], Ursula Panzner[1], Ligia Maria Cruz Espinoza[1], Vera von Kalckreuth[1], Justin Im[1], Heidi Schütt-Gerowitt[1,6], John A. Crump[7,8,9,10], Robert F. Breiman[11,12], Yaw Adu-Sarkodie[13,14], Ellis Owusu-Dabo[13,14], Raphaël Rakotozandrindrainy[15], Abdramane Bassiahi Soura[16], Abraham Aseffa [17], Nagla Gasmelseed[18,19], Karen H. Keddy[20,21], Jürgen May [22,23], Amy Gassama Sow[24,25], Peter Aaby[26,27], Holly M. Biggs[7,8], Julian T. Hertz[7,8], Joel M. Montgomery[11], Leonard Cosmas[11], Beatrice Olack[28], Barry Fields[11], Nimako Sarpong[13,23], Tsiriniaina Jean Luco Razafindrabe[15], Tiana Mirana Raminosoa[15], Leon Parfait Kabore[29], Emmanuel Sampo[29], Mekonnen Teferi[17], Biruk Yeshitela[17], Muna Ahmed El Tayeb[18], Arvinda Sooka[20], Christian G. Meyer[30,31], Ralf Krumkamp[22], Denise Myriam Dekker[22,23], Anna Jaeger[22], Sven Poppert[32], Adama Tall[25], Aissatou Niang[25], Morten Bjerregaard-Andersen[26,27], Sandra Valborg Løfberg[26,27], Hye Jin Seo[1], Hyon Jin Jeon[1], Jessica Fung Deerin[1], Jinkyung Park[1], Frank Konings[1], Mohammad Ali [1,33], John D. Clemens[1,34,35], Peter Hughes[36], Juliet Nsimire Sendagala[36], Tobias Vudriko[36], Robert Downing[37,38], Usman N. Ikumapayi[39], Grant A. Mackenzie[39,40,41], Stephen Obaro[42,43,44], Silvia Argimon[4], David M. Aanensen[4,45], Andrew Page[4], Jacqueline A. Keane[4], Sebastian Duchene[46], Zoe Dyson [46], Kathryn E. Holt [46], Gordon Dougan[4,47], Florian Marks [1,47] & Stephen Baker[2,3,47]

[1]International Vaccine Institute, SNU Research Park, 1 Gwanak-ro, 1 Gwanak-gu, Seoul 08826, Republic of Korea. [2]The Hospital for Tropical Diseases, Wellcome Trust Major Overseas Programme, Oxford University Clinical Research Unit, 764 Vo Van Kiet, Quant 5, Ho Chi Minh City, Vietnam. [3]Centre for Tropical Medicine and Global Health, University of Oxford, Old Road campus, Roosevelt Drive, Headington, Oxford OX3 7FZ, UK. [4]Centre for Genomic Pathogen Surveillance, Wellcome Genome Campus, Hinxton, Cambridge CB10 1SA, UK. [5]Addenbrooke's Hospital, Cambridge University Hospitals NHS Foundation Trust, Cambridge Biomedical Campus, Hills Road, Cambridge CB2 0QQ, UK. [6]Institute of Medical Microbiology, University of Cologne, 50923 Cologne, Germany. [7]Kilimanjaro Christian Medical Centre, P.O. Box 3010 Moshi, Tanzania. [8]Division of Infectious Diseases and International Health, Duke University Medical Center, Durham, NC 27710, USA. [9]Duke Global Health Institute, Duke University, Durham, NC 27708, USA. [10]Centre for International Health, University of Otago, Dunedin 9054, New Zealand. [11]Centers for Disease Control and Prevention, KEMRI Complex, Mbagathi Road off Mbagathi Way, P.O. Box 606-00621 Village Market, Nairobi, Kenya. [12]Emory Global Health Institute, Emory University, 1599 Clifton Road, NE, Atlanta, GA 30322, USA. [13]Kwame Nkrumah University of Science and Technology, P.O. Box PMB KNUST, Kumasi, Ghana. [14]Kumasi Centre for Collaborative Research in Tropical Medicine, Kwame Nkrumah University of Science and Technology, KCCR, UPO, PMB, KNUST, Kumasi, Ghana. [15]University of Antananarivo, BP 566 Antananarivo 101, Madagascar. [16]Institut Supérieur des Sciences de la Population, University of Ouagadougou, 03 B.P. 7118 Ouagadougou 03, Burkina Faso. [17]Armauer Hansen Research Institute, Jimma Road, ALERT Compound  P.O. Box 1005 Addis Ababa, Ethiopia. [18]Faculty of Medicine, University of Gezira, P.O. Box 20, Wad Medani, Sudan. [19]Faculty of Science, University of Hafr Al Batin, Al Jamiah, Hafr Albatin 39524, Saudi Arabia. [20]National Institute for Communicable Diseases, Private Bag X4, Sandringham, 2131 Johannesburg, South Africa. [21]Faculty of Health Sciences, University of the Witwatersrand, 1 Jan Smuts Avenue, Braamfontein, 2000 Johannesburg, South Africa. [22]Bernhard Nocht Institute for Tropical Medicine, Bernhard Nocht Str. 74, 20359 Hamburg, Germany. [23]German Center for Infection Research, Hamburg-Borstel-Lübeck, Inhoffenstrabe 7, 38124 Braunschweig, Germany. [24]Institute Pasteur de Dakar, 36 Avenue Pasteur, B.P. 220 Dakar, Senegal. [25]Université Cheikh Anta Diop de Dakar, Dakar B.P. 5005, Senegal. [26]Bandim Health Project, Apartado, 8611004 Bissau codex, Bissau, Guinea-Bissau. [27]Research Center for Vitamins and Vaccines, Bandim Health Project, Statens Serum Institut, 5 Artillerivej, DK-2300 Copenhagen, Denmark. [28]Kenya Medical Research Institute, Mbagathi Rd. P.O. BOX 54840-00200, Nairobi, Kenya. [29]Schiphra Hospital, Ouagadougou 01 01 B.P. 121, Burkina Faso. [30]Institute of Tropical Medicine, Eberhard-Karls University Tübingen, Geschwister-Scholl-Platz, 72074 Tübingen, Germany. [31]Duy Tan University, 254 Nguyen Van Linh, Da Nang, Vietnam. [32]Swiss Tropical and Public Health Institute, University of Basel, Socinstrasse 57, Postfach, CH-4002 Basel, Switzerland. [33]Johns Hopkins Bloomberg School of Public Health, 615 N Wolfe St., Baltimore, MD 21205, USA. [34]International Centre for Diarrheal Disease Research, Bangladesh (ICDDR,B), GPO Box 128Dhaka 1000, Bangladesh. [35]University of California, Fielding School of Public Health, 650 Charles E. Young Dr. South, Center for Health Sciences, Los Angeles, CA 90095-1772, USA. [36]Medical Research Center (MRC)/Uganda Virus Research Institute (UVRI) & London School of Hygiene and Tropical Medicine (LSHTM) Uganda Research Unit, Plot 51-59 Nakiwogo Road P.O. Box 49Entebbe, Uganda. [37]Uganda Virus Research Institute (UVRI), Plot 51-59 Nakiwogo Road P.O. Box 49Entebbe, Uganda. [38]Public Health Emergency Operations Centre, Plot 6 Lourdel Road, Nakasero PO Box 7272, Kampala, Uganda. [39]Medical Research Council Unit The Gambia at the London School of Hygiene and Tropical Medicine, Atlantic Boulevard, Fajara, P.O. Box 273, Banjul, The Gambia. [40]Murdoch Children's Research Institute, 50 Flemington Road Parkville Victoria, 3052 Melbourne, Australia. [41]Institut de Recherche en Santé, de Surveillance Epidemiologique et de Formations (IRESSEF), Arrondissement 4 Rue 2D1, BP 7325, Dakar, Senegal. [42]Division of Pediatric Infectious Diseases, University of Nebraska Medical Center, 42nd and Emile, Omaha, Nebraska 68198, USA. [43]University of Abuja Teaching Hospital, Gwagwalada, P.M.B 228, Abuja, Nigeria. [44]Bingham University, P.M.B 005, KM 26 Abuja-Keffi Expressway Kodope, Karu, Nasarawa State, Nigeria. [45]Big Data Institute, University of Oxford, Old Road Campus, Oxford OX3 7LF, UK. [46]Department of Biochemistry and Molecular Biology, Bio21 Molecular Science and Biotechnology Institute, University of Melbourne, 30 Flemington Road, Parkville, Victoria 3010, Australia. [47]The Department of Medicine, The University of Cambridge, Box 157, Hills Road, Cambridge CB2 0QQ, UK. These authors contributed equally: Florian Marks, Stephen Baker

