## [Peer Review File · Nature Communications]

Reviewers' comments:

Reviewer #1 (Remarks to the Author):

This important paper is focused on the newly-discovered typhoid epidemic in sub-Saharan Africa, and focuses on 135 isolates of *S. Typhi* from nine African countries, put into the context of about 600 African *S. Typhi* genomes. It presents an authoritative description of the basis of the MDR typhoid fever that was found in the TSAP study, and addresses phylogeographic aspects of the epidemic.

This is an excellent piece of work, that has been written extremely well. I just have a number of specific comments:

Line 203 - The phrase "not providing any evidence for recent inter-continental transmission as observed for 4.3.1" is not clear and should be rephrased.

Line 208 - Discuss how the timings derived from the BEAST2 analysis correspond with the timing of the Typhoid epidemics in each country.

Line 255 - Define "PYO".

Line 258- rephrase "though some isolates" to clarify this point.

Line 286- change to "...antimicrobial usage in west Africa".

Line 288- Rephrase "these same *gyrA* mutations were not found in..." and "MDR against alternative antimicrobial agents" to improve clarity.

Line 290- explain what the first-line antibiotics are.

Line 297- this important point should be emphasised more in the paper, and properly discussed. This is an important difference to the observation that was recently made by Kate Baker (doi: 10.1038/s41467-018-03949-8), who discovered that AMR plasmids are routinely transferred between *Shigella* isolates in the context of a serious outbreak.

Line 298- the authors do not present sufficient data to justify this statement, and should moderate the text. While being true that they do not find Plasmid-associated sequences to definitely state the the IncH Plasmid was not present would require support from PacBio or other long-Read sequencing. On Line299, change "signifying" to "suggesting".

Line 328- more data are required to definitively state that the AMR genes are "both Plasmid and chromosomally located", which would require support from PacBio or other long-read sequencing technique.

Line 331- Rephrase "is now characterised ciprofloxacin resistance".

Line 337- rather than cite Ref 27, refer to the recent Klemm et al paper, published in mBio (doi: 10.1128/mBio.00105-18).

Line 338- rephrase "though not exhibited in our analysis".

Line 350- Rephrase, to avoid using the words "these" and "their".

Line 401- I applaud the authors for making their data available as an online Microreact project. To make this important resource more obvious to the reader, the URL should also be mentioned elsewhere - either in the Abstract, or at the end of the Introduction.

Line 415- Rephrase: "is it incurs..."

Line 417- add a few more words to explain what being "sensitive to the choice of prior distribution" actually means.

Line 461-462 - sentence is unclear, the reader may not know what "adjustment factors" are.

Figure 1 - Rather than (or as well as) showing the "substitutions per site" scale bar, add a scale bar that shows the number of core genome SNPs. Perhaps a 5 or 10 SNP scale bar would be appropriate?

Figure 2 - the Sudan data look odd, were there really Zero isolates? Does not make sense...

Figure 3 - the colours in Figure 3b need to be improved. It is difficult to reword the branches of the tree to individual countries because 4 different types of Green have been used. Please broaden the colour palette! The colours for each country should be the same as used in Figure 4.

Figure S2- in the legend, what does the term "correct sampling times" mean? Rephrase.

Figure S3- define LSD.

Reviewer #2 (Remarks to the Author):

Overview:

The authors sought to undertake a substantial study looking at drug resistant typhoid fever in sub-Saharan Africa. A previous study by the main authors on typhoid fever in Asia forms the prelude to this study, which comes across as the next-in-line 'Africa' project. This is a useful descriptive study, but it is not of sufficiently broad interest and the methods and findings are not suitably novel for publication in Nature Communications.

Too little space has been given to describing the typhoid isolates upon which the study is based. There is no indication of whether isolates are epidemiologically linked by transmission or part of an outbreak. If this data is unavailable, it should at least be acknowledged in the discussion as a possibility. The authors do say that higher MDR levels are seen in countries with the higher burden of typhoid, but then go on to draw conclusions about the lack of MDR in lower burden countries where, in many cases, the number of *S. Typhi* isolates was below 10.

Small issues:

Line 128: 'hamper' instead of 'hampers'

Line 136: 'towards this group' makes more sense than 'against'

Line 176: query whether 52% is 'highly prevalent'

Line 221: genes 'catalysing' resistance ??

Line 255: define PYO on first use

Line 521: Reference 12 and 13 are the same paper

Larger queries:

Figure 2: The colouring is unclear. Recommend switching the white and grey to make it obvious

which countries have been sampled.

Figure 3b: The colour is very unclear, making interpretation of the figure impossible. The results given in the text are very interesting, so the figure really needs to back this up.

Line 190: I would question that the Kenyan isolates form 'distinct clades'. Figure 3a shows 3 isolates in lineage II, and the Kenyan isolates are interspersed with Tanzanian ones in lineage I. As the MDR isolates are not highlighted, this statement is impossible to verify.

Line 198: Is clonal expansion the only possibility? What about an outbreak?

Line 336-338. If this is not exhibited in the authors' analysis, then further weight needs to be given to this speculation. Or it should be removed.

Line 340: Are these 'key locations' simply the ones in the study? In which case, what makes them key? If not, the authors should expand upon which are the key locations.

Methods: 95% CIs for incidence values were calculated according to the methods section. Why were they not included in the text of the results.

Reviewer #3 (Remarks to the Author):

Reviewed by Dr Tom Connor, Cardiff University

General Comments

This is an interesting, relevant paper that provides a novel set of findings (particularly around clade 3.1.1 and the patterns of MDR of Typhi in sub-saharan Africa) which would, in my view, be of broad interest to others in the community and the wider field.

From a technical standpoint the bioinformatics/population genetics approaches used all seem appropriate, and the methods have a good level of detail such that I am pretty sure that I could reproduce this work without an issue.

I have a number of questions/comments, which come not from the fact that there are issues with the paper as it stands, but due to the fact that I think that there is a lot of really interesting and valuable data here, which could be enhanced through considering a couple of questions that would add to results discussed in the text.

Minor comments/questions

While I think the paper is interesting, novel and of interest to the wider audience, I had a few questions/comments that the authors could consider, the answers to which may add value/improve the paper.

The paper is generally well written, but in a few places, the terminology used is a little confusing, or could be clearer. For example, on line 187 the authors refer to 'distinct origins' of the MDR phenotypes in each region. I don't think the meaning of this is quite clear. Similarly, on the next line, the authors mention 'novel genome sequences' - I presume these are the samples that have been sequenced for the study, but it seemed an odd way to describe those samples. Collectively, I think the readability could probably be improved a bit by simplifying some of this terminology, or dropping some unnecessary adjectives (such as 'novel') and double checking the usage of the term 'clonal expansion', as I know that this can make some microbiologists angry. On line 196 the

phrase 'not closely related' is used to describe relationship to other lineage II organisms, without an indication of what one might consider close, or quantitatively how far apart they are. Have you performed any sort of BEAST analysis for this part of the dataset? an MRCA for those sublineages, and collectively for lineage II would cover this aspect off nicely.

On line 197/198 the paper states that a mean pairwise distance of 1 SNP in the Ugandan isolates is evidence of a clonal expansion. I don't think that this is necessarily true. I think the shape of the tree suggests that it is some sort of expansion, but the low level of diversity across the cluster is, more saliently, evidence of a recent introduction, followed by (possibly rapid) expansion. I suspect with a BEAST run may become clearer/be more easily presented, and would allow the authors to be able to make a slightly more explicit, more accurate, and arguably stronger statement. Again reading along to line 199, a BEAST analysis would be nice if possible, as the question in my mind was when did the likely introduction of the Ugandan isolates happen, and how far apart (in terms of time/SNPs) are they from an MRCA with the Asian isolates they cluster with.

For the part from line 202 on 3.1.1, you don't mention numbers for the countries involved, as you had done for the preceding part of this section of the paper. This would be useful (particularly for the Ghana isolates) and would be consistent with the way the first part of the section was presented. Figure 3b is really excellent, and tells the story quite beautifully.

Around line 255, there are interesting discussions around the plasmids that are present, and that form the basis for the MDR phenotype. I have two questions, firstly, how well does the plasmid phylogeny correspond to the phylogenetic trees? i.e. can the authors gain an indication of the likely number of acquisition events that have occurred, and, if so, can this be linked to the BEAST data to infer when plasmids were acquired?

My last question/comment is around the differences in rates of MDR by age group. I wondered if the authors had any idea why this might be, and if they noticed any correlations of MDR rates with other public health issues such as rates of HIV or TB infection?

Reviewer #1 (Remarks to the Author):

This important paper is focused on the newly-discovered typhoid epidemic in sub-Saharan Africa, and focuses on 135 isolates of *S. Typhi* from nine African countries, put into the context of about 600 African *S. Typhi* genomes. It presents an authoritative description of the basis of the MDR typhoid fever that was found in the TSAP study, and addresses phylogeographic aspects of the epidemic.

This is an excellent piece of work, that has been written extremely well. I just have a number of specific comments:

We thank the reviewer for their comments and the acknowledgement of the work.

Line 203 - The phrase “not providing any evidence for recent inter-continental transmission as observed for 4.3.1” is not clear and should be rephrased.

This has been rephrased to “showing no evidence for inter-continental transmission as observed for 4.3.1”.

Line 208 - Discuss how the timings derived from the BEAST2 analysis correspond with the timing of the Typhoid epidemics in each country.

Outbreaks of typhoid fever in West Africa are under-reported and publications on typhoid epidemics in this region are limited. We assumed the majority of the countries in West Africa represented in the BEAST2 analysis, such as Ghana and Burkina Faso, are typhoid endemic countries.

Line 255 - Define “PYO”.

PYO defined as person years of observation (PYO).

Line 258- rephrase “though some isolates” to clarify this point.

Revised to “no MDR S. Typhi were detected; 2/14 isolates were resistant to chloramphenicol and co-trimoxazole”.

Line 286- change to “...antimicrobial usage in west Africa”.

Revised to “antimicrobial usage in West Africa”.

Line 288- Rephrase “these same gyrA mutations were not found in...” and “MDR against alternative antimicrobial agents” to improve clarity.

Response: Rephrased to “Conversely, no gyrA mutations were found in the MDR S. Typhi 3.1.1 from Ghana.”

Line 290- explain what the first-line antibiotics are.

Rephrased to “first-line antimicrobial agents (ampicillin, chloramphenicol, and co-trimoxazole)”.

Line 297- this important point should be emphasised more in the paper, and properly discussed. This is an important difference to the observation that was recently made by Kate Baker (doi: 10.1038/s41467-018-03949-8), who discovered that AMR plasmids are routinely transferred between Shigella isolates in the context of a serious outbreak.

Rephrased to “The distinct MDR lineages of S. Typhi found in West and East Africa, each associated with a distinct IncH11 plasmid sequence type, suggest that S. Typhi and its AMR plasmids have not been transferred laterally across the continent. This may be because genotype 4.3.1 MDR S. Typhi has not been circulating for a sufficient period in Africa to reach the West African region.”

Line 298- the authors do not present sufficient data to justify this statement and should moderate the text. While being true that they do not find Plasmid-associated sequences to definitely state the IncH Plasmid was not present would require support from PacBio or other long-Read sequencing.

Rephrased to “Furthermore, the four MDR S. Typhi isolates from Tanzania did not harbour plasmid-associated sequences, suggesting that these AMR genes are inserted into the chromosome as has been observed previously in Asia”.

Line 242 has also been changed: “Additionally, none of the four MDR organisms from Tanzania possessed a detectable plasmid backbone. Using Bandage to investigate the location of MDR cassettes, we found that these isolates carried multiple resistance genes”

On Line299, change “signifying” to “suggesting”.

Revised to “suggesting”.

Line 328- more data are required to definitively state that the AMR genes are “both Plasmid and chromosomally located”, which would require support from PacBio or other long-read sequencing technique.

We investigated the locations of these MDR cassettes using Bandage as mentioned in line 242 (result section).

Line 331- Rephrase “is now characterised ciprofloxacin resistance”.

Rephrased to “Typhi in Africa. 4.3.1 S. Typhi has spread successfully cross South Asia and become increasingly resistant to ciprofloxacin, making treatment options more limited”

Line 337- rather than cite Ref 27, refer to the recent Klemm et al paper, published in mBio (doi: 10.1128/mBio.00105-18).

Ref 27 has been corrected to Klemm et al. paper as suggested.

Line 338- rephrase “though not exhibited in our analysis”.

Rephrased to “South Africa suggests possible spread from East Africa to Southern Africa through human population movement, however this notion requires further investigation.”

Line 350- Rephrase, to avoid using the words “these” and “their”.

These have been changed where appropriate

Line 401- I applaud the authors for making their data available as an online Microreact project. To make this important resource more obvious to the reader, the URL should also be mentioned elsewhere - either in the Abstract, or at the end of the Introduction.

We have included the URL in the abstract as suggested

Line 415- Rephrase: “is it incurs...”

Rephrased to “A potential shortcoming of this analysis was that it includes a large number of parameters (transition rates between all locations)”.

Line 417- add a few more words to explain what being “sensitive to the choice of prior distribution” actually means.

Changed to “therefore the output of these analysis may be affected by the prior distribution.”

Line 461-462 - sentence is unclear, the reader may not know what “adjustment factors” are.

Rephrased to “The incidence of MDR S. Typhi in Uganda could not be measured, as data regarding adjustment factors (healthcare seeking behaviour and recruitment proportion) was unavailable at the time of analysis.” though these are mentioned in the preceding sentences of this paragraph on method of incidence calculation.

Figure 1 - Rather than (or as well as) showing the “substitutions per site” scale bar, add a scale bar that shows the number of core genome SNPs. Perhaps a 5 or 10 SNP scale bar would be appropriate?

Response: We are not trying to interpret the SNP distances between the genotypes in Figure 1, the existing scale bar with substitutions per site generated by maximum likelihood method. Therefore, the scale bar on the Figure 1 is equivalent to 264 SNPs, and would be difficult to show a 5-10 SNP scale bar.

Figure 2 - the Sudan data look odd, were there really Zero isolates? Does not make sense...

There were zero cases from Sudan in our TSAP study findings (reported in Marks et al, Lancet Glob Heal 2017; 5: e310–23).

Figure 3 - the colours in Figure 3b need to be improved. It is difficult to reword the branches of the tree to individual countries because 4 different types of Green have been used. Please broaden the colour palette! The colours for each country should be the same as used in Figure 4.

Response: These colours in the figures have been revised

Figure S2- in the legend, what does the term “correct sampling times” mean? Rephrase.

Rephrased to “The black symbol highlights actual sampling time”

Figure S3- define LSD.

Rephrased to “The red lines represent the estimated Least-squares dating (LSD)”

Reviewer #2 (Remarks to the Author):

Overview:

The authors sought to undertake a substantial study looking at drug resistant typhoid fever in sub-Saharan Africa. A previous study by the main authors on typhoid fever in Asia forms the prelude to this study, which comes across as the next-in-line ‘Africa’ project. This is a useful descriptive study, but it is not of sufficiently broad interest and the methods and findings are not suitably novel for publication in Nature Communications.

Many thanks for your comments and opinion, we have tried to address these accordingly.

Too little space has been given to describing the typhoid isolates upon which the study is based. There is no indication of whether isolates are epidemiologically linked by transmission or part of an outbreak. If this data is unavailable, it should at least be acknowledged in the discussion as a possibility. The authors do say that higher MDR levels are seen in countries with the higher burden of typhoid, but then go on to draw conclusions about the lack of MDR in lower burden countries where, in many cases, the number of S. Typhi isolates was below 10.

A description of the isolates collected and analysed in this manuscript is included in the method, introduction, and abstract sections (with a reference to relevant publication that elaborates our sample collection methods). For the TSAP study, as the surveillance duration lasted for several years, any outbreak events would have been detected, and study sites with high a typhoid burden and high MDR typhoid were typhoid endemic settings.

The burden of typhoid (with respect to MDR typhoid calculation) in the respective countries was based on the adjusted incidences of typhoid fever in person years of observation (published by Marks et al in Lancet Global Health). The crude case numbers of S. Typhi isolates were adjusted by the healthcare seeking behaviour at the respective study site and recruitment of study eligible patients. We did not make any conclusions about the lack of MDR in lower burden countries. Alternatively, we state that a high incidence of MDR typhoid was found in countries with a high burden of typhoid fever (in lines 306-309). We found no MDR S. Typhi in low burden countries. We also concluded that the incidence of MDR typhoid may vary between settings and is associated with differing exposures to antimicrobials (in lines 321-333).

Small issues:

Line 128: ‘hamper’ instead of ‘hampers’

This has been revised

Line 136: 'towards this group' makes more sense than 'against'

Revised to "susceptibility to this group of antimicrobials". (Did not replace with "toward" since this vocabulary is already used at the beginning of this sentence.)

Line 176: query whether 52% is 'highly prevalent'

"Highly" has been replaced with "MDR phenotype was prevalent across..."

Line 221: genes 'catalysing' resistance ??

Revised "genes catalyzing resistance to" to "genes encoding resistance to".

Line 255: define PYO on first use

Revised to "person years of observation (PYO)".

Line 521: Reference 12 and 13 are the same paper

These have been rectified

Larger queries:

Figure 2: The colouring is unclear. Recommend switching the white and grey to make it obvious which countries have been sampled.

This has been edited

Figure 3b: The colour is very unclear, making interpretation of the figure impossible. The results given in the text are very interesting, so the figure really needs to back this up.

The colours have been revised accordingly

Line 190: I would question that the Kenyan isolates form 'distinct clades'. Figure 3a shows 3 isolates in lineage II, and the Kenyan isolates are interspersed with Tanzanian ones in lineage I. As the MDR isolates are not highlighted, this statement is impossible to verify.

The 3 Kenyan isolates in lineage II (yielded from our study) clustered with the previous collection of isolates from Kenya as shown in this Figure 3a (bar).

Another bar has been added in Figure 3a to highlight the MDR and non-MDR isolates.

The sentence has been rephrased to: “Our Kenyan MDR 4.3.1 organisms (2012-2013) belonged to two distinct clades”

Line 198: Is clonal expansion the only possibility? What about an outbreak?

Rephrased to “indicative of a recent population expansion or an outbreak.¹⁶” and added Ref 16 (Kabwama SN et al; BMC Public Health 2017; 17: 1-9) on the 2015 typhoid fever outbreak in Kampala, Uganda.

Line 336-338. If this is not exhibited in the authors’ analysis, then further weight needs to be given to this speculation. Or it should be removed.

Rephrased to “Further, the emergence of MDR S. Typhi 4.3.1 in South Africa suggests possible spread from East Africa to Southern Africa through human population movement, however this notion requires further investigation.²²”

Line 340: Are these ‘key locations’ simply the ones in the study? In which case, what makes them key? If not, the authors should expand upon which are the key locations.

Key locations referred to high MDR prevalence. The sentence has been rephrased to “This study highlights locations in sub-Saharan Africa where MDR typhoid is prevalent and where future activities to control its spread from Asia into Africa and also within Africa could be focused”

Methods: 95% CIs for incidence values were calculated according to the methods section. Why were they not included in the text of the results.

They were not included due to the word limit. However, all the 95% CIs are now included in the text of the results.

Reviewer #3 (Remarks to the Author):

Reviewed by Dr Tom Connor, Cardiff University

General Comments

This is an interesting, relevant paper that provides a novel set of findings (particularly around clade 3.1.1 and the patterns of MDR of Typhi in sub-saharan Africa) which would, in my view, be of broad interest to others in the community and the wider field.

From a technical standpoint the bioinformatics/population genetics approaches used all seem appropriate, and the methods have a good level of detail such that I am pretty sure that I could reproduce this work without an issue.

I have a number of questions/comments, which come not from the fact that there are issues with the paper as it stands, but due to the fact that I think that there is a lot of really interesting and valuable data here, which could be enhanced through considering a couple of questions that would add to results discussed in the text.

Many thanks for your input and comments, these have been addressed below

Minor comments/questions

While I think the paper is interesting, novel and of interest to the wider audience, I had a few questions/comments that the authors could consider, the answers to which may add value/improve the paper.

The paper is generally well written, but in a few places, the terminology used is a little confusing, or could be clearer. For example, on line 187 the authors refer to 'distinct origins' of the MDR phenotypes in each region. I don't think the meaning of this is quite clear. Similarly, on the next line, the authors mention 'novel genome sequences' - I presume these are the samples that have been sequenced for the study, but it seemed an odd way to describe those samples. Collectively, I think the readability could probably be improved a bit by simplifying some of this terminology, or dropping some unnecessary adjectives (such as 'novel') and double checking the usage of the term 'clonal expansion', as I know that this can make some microbiologists angry. On line 196 the phrase 'not closely related' is used to describe relationship to other lineage II organisms, without an indication of what one might consider close, or quantitatively how far apart they are. Have you performed any sort of BEAST analysis for this part of the dataset? an MRCA for those sublineages, and collectively for lineage II would cover this aspect off nicely.

We have rephrased and removed the suggested words where appropriate

On line 197/198 the paper states that a mean pairwise distance of 1 SNP in the Ugandan isolates is evidence of a clonal expansion. I don't think that this is necessarily true. I think the shape of the tree suggests that it is some sort of expansion, but the low level of diversity across the cluster is, more saliently, evidence of a recent introduction, followed by (possibly rapid) expansion. I suspect with a BEAST run may become clearer/be more easily presented, and would allow the authors to be able to make a slightly more explicit, more accurate, and

arguably stronger statement. Again reading along to line 199, a BEAST analysis would be nice if possible, as the question in my mind was when did the likely introduction of the Ugandan isolates happen, and how far apart (in terms of time/SNPs) are they from an MRCA with the Asian isolates they cluster with.

We have rephrased the sentence as “indicative of a recent population expansion or an outbreak.¹⁶” BEAST could not be used on data from this population because there was no temporal structure within this population. Therefore, a molecular clock could not be trained to these data. The use of BEAST, therefore would be inappropriate.

For the part from line 202 on 3.1.1, you don't mention numbers for the countries involved, as you had done for the proceeding part of this section of the paper. This would be useful (particularly for the Ghana isolates) and would be consistent with the way the first part of the section was presented. Figure 3b is really excellent, and tells the story quite beautifully.

Rephrased and added a reference to Table 1. “In contrast, the 3.1.1 MDR S. Typhi from Ghana (68 isolates) represented a population that was found only in West Africa, with the resulting phylogeny showing no evidence for inter-continental transmission as observed for 4.3.1. (Table 1)”

Around line 255, there are interesting discussions around the plasmids that are present, and that form the basis for the MDR phenotype. I have two questions, firstly, how well does the plasmid phylogeny correspond to the phylogenetic trees? i.e. can the authors gain an indication of the likely number of acquisition events that have occurred, and, if so, can this be linked to the BEAST data to infer when plasmids were acquired?

We ran the phylogeny for the plasmids and found that the IncHII plasmid sequence type 2a was identical in all Ghanaian isolates. The maximum likelihood phylogenetic tree and the BEAST phylogenetic tree of genotype 3.1.1 suggested that there may have been multiple acquisition events of this plasmid across different sublineages, or a single acquisition event followed by plasmid transfer between sublineages. The earliest hypothetical ancestor of our MDR isolates existed around 2000, so this may be the time of the first MDR plasmid acquisition.

My last question/comment is around the differences in rates of MDR by age group. I wondered if the authors had any idea why this might be, and if they noticed any correlations of MDR rates with other public health issues such as rates of HIV or TB infection?

We did not perform any analyses on any correlations of MDR typhoid rates with HIV or TB as these tests were not included in our study. Our speculative thoughts on this include potentially differing exposures to antimicrobials in different age groups and settings. This is noted in the discussion section.

REVIEWERS' COMMENTS:

Reviewer #3 (Remarks to the Author):

I thank the authors for their considered responses to the reviews and edits to the manuscript. I think the paper is now clearer than it was, and reads very well.

I have no further comments or suggestions for changes.

NCOMMS-18-13127B

Response to Referees:

Reviewer #3 (Remarks to the Author):

I thank the authors for their considered responses to the reviews and edits to the manuscript. I think the paper is now clearer than it was, and reads very well.

I have no further comments or suggestions for changes.

We thank the reviewer for their comments and the acknowledgement of the work once again.